# The α5-α6-α7-Pba3-Pba4 Complex: A Starting Unit in Proteasome Core Particle Assembly

**DOI:** 10.3390/biom15050683

**Published:** 2025-05-08

**Authors:** Ana C. Matias, Margarida N. Tiago, Jessica Zimmermann, R. Jürgen Dohmen, Paula C. Ramos

**Affiliations:** 1Center of Molecular Biosciences, Institute for Genetics, Department of Biology, Faculty of Natural Sciences and Mathematics, University of Cologne, 50674 Cologne, Germany; ana.matias@ipma.pt (A.C.M.);; 2Departamento de Química e Bioquímica, Faculdade de Ciências e Tecnologia, Universidade do Algarve, 8000-117 Faro, Portugal

**Keywords:** proteasome core particle, assembly, chaperones, Pba3-Pba4, Blm10

## Abstract

A complex composed of Pba3-Pba4 and subunits α5, α6, and α7 is identified as an early intermediate in proteasome core particle assembly in wild-type *Saccharomyces cerevisiae* cell*s*. The same complex can be reconstituted from recombinantly produced components in vitro. Assembly of α6 and α7 with Pba3-Pba4 depends on the presence of the α5 subunit, the binding of which apparently initiates the formation of this intermediate. Our data suggest the following order of events: first, Pba3-Pba4 binds α5, then α6 is incorporated, and at the end α7. In the absence of the chaperones Pba1-Pba2 or Ump1, alternative Pba4-containing complexes are detected, the formation of which depends on the Blm10/PA200 protein. Overexpression of Pba1-Pba2 abolishes the formation of these complexes containing Pba4 and Blm10, suggesting that Blm10 may replace Pba1-Pba2 as an alternative assembly factor.

## 1. Introduction

The 26S proteasome of eukaryotic cells is a very large protein complex (>2 MDa), the principal structure of which is highly conserved from yeast to humans [1]. As a central component of the ubiquitin/proteasome system (UPS), it serves essential functions in protein homeostasis and the control of many cellular processes [2]. The proteasome is composed of a 20S core particle (CP) and two 19S regulatory particles (RP) [1,2]. These subcomplexes assemble largely independently of each other with the support of dedicated assembly chaperones [1,3]. The 20S CP, in its mature form, consists of two outer rings of seven α-subunits and two inner rings of β-subunits [4]. CP biogenesis is promoted by three dedicated assembly chaperones (Ump1, Pba1-Pba2, Pba3-Pba4), as well as general chaperones of the Hsp70 and Hsp110 family [1,3,5].

In mammalian cells, 20S proteasome core particles (CP) were suggested to assemble via α-rings [6]. Besides being composed of seven distinct α-subunits, such rings, which were observed upon knockdown of *UMP1* or β subunit expression, were characterized by the presence of two chaperone pairs PAC1-PAC2 and PAC3-PAC4 [7,8]. Orthologues of these assembly chaperones involved in CP biogenesis were found in *S. cerevisiae* and called Pba1-Pba2 and Pba3-Pba4, respectively [3,6]. However, an α-ring intermediate has not been detected in this yeast thus far [5]. The goal of our study, therefore, was to characterize early steps in CP biogenesis with a focus on the roles of Pba3-Pba4, and to determine whether this chaperone coexists with Pba1-Pba2 in intermediates such as α-rings.

A crystal structure of Pba3-Pba4 in complex with the proteasome α5 subunit has been reported [9]. Modeling the described structure into the available CP structures, utilizing the α5 subunit for the alignment of both structures, revealed that the Pba3-Pba4 dimer would be in the interface between the α and β rings, on top of the α5 subunit, with Pba4 pointing towards α4 and Pba3 contacting α6. According to this modeling, Pba3-Pba4 would extend into the inner chamber of the CP, approximately located where subunits β5 and β6 would be in the mature CP [9]. In a pulldown assay, followed by mass spectrometry, α6 was also identified as a subunit interacting with Pba3-Pba4. Further functional characterization revealed that Pba3-Pba4 is required for the proper incorporation of the α4 subunit into CP α-rings [9]. Another study concluded that this heterodimeric chaperone, by binding to specific α-subunits, guarantees the incorporation of α3 precisely between α2 and α4. In cells lacking Pba3-Pba4, CP α-rings with a second copy of α4 instead of α3 were detected [10]. Independent evidence led to the conclusion that Pba3-Pba4 promotes the interaction between α4 and α5, and that in its absence, α2 replaces α4 in the α-ring [11]. Based upon these discoveries, it can be concluded that without Pba3-Pba4, the CP pool in *S. cerevisiae* cells is not homogeneous but rather bears proteasomes with different α-ring compositions and organization [10]. A similar conclusion was reached for the formation of alternative CP forms occurring in the absence of PAC3-PAC4 in human cells, indicating functional conservation of this assembly chaperone from yeast to humans. How exactly this conserved chaperone acts in early assembly steps, and thereby promotes efficient and accurate biogenesis of CPs, is not well understood. Our characterization of Pba3-Pba4-containing complexes showed that, in wild-type yeast cells, this chaperone dimer associates with subunits α5, α6, and α7. Consistent with these findings, we were able to recapitulate the formation of the Pba3-Pba4α5-α6-α7 complex in vitro. Additionally, in the absence of a functional Pba1-Pba2 or Ump1, we detected Pba4 in complexes containing Blm10.

## 2. Materials and Methods

### 2.1. Yeast Strains, Cultures, and Media

Yeast rich (YPD) and synthetic minimal media with 2% dextrose were prepared as described [12]. *S. cerevisiae* cells from exponentially growing cultures (OD_600_ 0.8–1.2) were harvested at 3000× *g* rcf, washed with water, frozen in liquid nitrogen, and stored at −80 °C. For gel filtration experiments, 200 mL culture volume was used; for native gel experiments, only 15 mL. For purification of complexes, cells from up to 4 L of culture were used as starting material. All strains used are listed in Table 1. Construction of strains in which Pba4 was C-terminally tagged with 2xHA or FLAG-6xHis (FH) instead of its untagged counterpart was performed according to a plasmid integration strategy described previously [13]. A *pba1∆* cassette was obtained by amplifying the *HIS3* marker from pFA6a-His3MX6 [14]. The *pba1Δ::HIS3* strain (PR118) or the strain *blm10Δ::KanMX pba1Δ::HIS3* (PR207) were prepared by gene replacement in the strains JD47-13C or *blm10Δ::KanMX*, respectively. Strains in which the expression of α or β subunit-encoding genes could be shut down (SD) by the addition of glucose were prepared by exchanging native promoters with P*_GALS_* or P*_GAL1_* [15]. Similar SD strains were generated for controlled expression of the *PBA1* and *PBA2* genes using P*_GAL1_*. To this end, pYM-N30 or pYM-N22 plasmids were employed as templates to generate PCR products that were used for genomic promoter transplacement, as described [16]. Correct insertion of P*_GALS/1_* in front of the desired genes was verified by analytical PCR.

To obtain conditional expression of *PBA4,* a plasmid was generated that contained a cassette enabling control of expression of *PBA4::HA* from P*_GAL1_* promoter and by three tetracycline-binding aptamers (tc elements) in the 5’ UTR [17]. This cassette was inserted into the 3’ UTR of the *URA3* gene. The entire unit was released from the plasmid by digestion with *NotI* and inserted into the genome by homologous recombination with the *ura3-52* locus selecting for uracil prototrophic transformants. All plasmids were verified by DNA sequencing, and correct yeast recombinants were identified by analytical PCR.

### 2.2. Protein Extraction, Gel Filtration, SDS- and Native PAGE, and Immunoblots

Total protein crude extracts were prepared in 26S buffer (50 mM Tris-HCl, pH 7.5, containing 1 mM DTT, 5 mM MgCl_2_, 2 mM ATP, 15% glycerol) [13]. For small culture volumes (15 mL), cells were lysed with glass beads (0.4–0.5 mm, Sigma-Aldrich Chemie GmbH, Taufkirchen, Germany) in extraction buffer by vortexing. For larger culture volumes, the cell paste was ground to powder with the help of a mortar in the presence of liquid nitrogen. Extraction buffer was added in a proportion of 2 mL/g of pelleted yeast cells. Cell debris was removed by centrifugation at 15,800× *g* rcf at 4 °C. The protein content in the supernatant was determined using the Bradford protein assay from Bio-Rad (Hercules, CA, USA). Extracts to be submitted to gel filtration fractionation were subjected to a second centrifugation at 60,000× *g* rcf for 30 min at 4 °C. Protein concentrations in extracts for parallel experiments were adjusted to 5 mg/mL using extraction buffer.

Extract samples containing 1 mg of total protein in a volume of 200 μL were fractionated on a Superdex200 column equilibrated with 26S buffer and coupled to an ÅKTA FPLC (GE Healthcare, Solingen, Germany). The flow rate was 0.35 mL/min, and fractions of 0.5 mL were collected. The Superdex200 column was calibrated using the following standards: ferritin (440 kDa), catalase (232 kDa), bovine serum albumin (67 kDa), and ovalbumin (43 kDa).

Native PAGE utilized Ready Gels (Tris-HCl 4–15% gradient gels; Bio-Rad). Samples were mixed with 4xNB (240 mM Tris pH 8.8, 80% (*v*/*v*) glycerol, 0.04% (*w*/*v*) bromophenol blue) and loaded onto the gel. Electrophoresis was performed in Laemmli running buffer without SDS at 16 mA for about 1.5 h with refrigeration. In vitro assembled complexes were resolved using pre-casted native gels (FastGene PAGE gels 4–20% from Nippon Genetics Europe, Düren, Germany). The gels were incubated for 10 min in transfer buffer containing 2% (*w*/*v*) SDS before electroblotting for 2 h at 0.8 mA/cm^2^. SDS-PAGE, membrane processing, and immuno-detection were performed as described previously [13], and the signals were detected using SuperSignal West Femto ECL reagent (Thermo Fisher Scientific, Karlsruhe, Germany). Antibodies used were: anti-HA (monoclonal rat 3F10-POD, Roche, Diagnostics, Mannheim, Germany), anti-FLAG (monoclonal mouse M2, Sigma-Aldrich Chemie GmbH, Taufkirchen, Germany), anti-α7 (polyclonal rabbit) [12], and anti-α5 as well as anti-α6 (polyclonal rabbit antisera raised against the subunits produced in *E. coli* and purified, as described in Section 2.4).

### 2.3. Immunoaffinity Purification of the Pba4-FLAG Complex

Cells from the strains PR211 or PR344 expressing Pba4-FH were grown at 30 °C in a culture volume of 2 L in a minimal medium without tryptophan until an OD_600_ of 1.2. Cells were pelleted and frozen in liquid nitrogen. The cells were reduced to powder with the help of a mortar in the presence of liquid nitrogen, and 50 mM Tris-HCl, pH 7.4 buffer containing 150 mM NaCl and protease inhibitors (Complete, EDTA-free, Roche Diagnostics, Mannheim, Germany) was added in the proportion of 1 mL/g wet weight. Cell debris was removed by centrifugation at 30,000× *g* rcf for 30 min at 4 °C. The supernatant was added to 0.5 mL Ni-NTA equilibrated in the same buffer and incubated for 1 h in batch with rotation at 4 °C. The eluted material was further incubated with 0.5 mL of anti-FLAG resin (equilibrated with the same buffer) for 1 h in batch with rotation at 4 °C. The suspension was transferred to an empty column. The flow-through was collected, and the FLAG resin was washed 3 times with 20 mL FLAG buffer. Bound proteins were eluted sequentially with 1 mL elution buffer containing FLAG peptide at concentrations of 50 µg/mL, 100 µg/mL, and 200 µg/mL. Proteins from the relevant steps were analyzed by SDS-PAGE and immunoblotting.

### 2.4. Expression of 8His-SUMO1-Fused Proteins in E. coli and Their Purification

Yeast α-subunits or Pba chaperones were expressed as fusions to 8His-tagged human SUMO1 from the plasmids listed in Table 2. Operon structures were generated to co-express subunits of heterodimeric chaperones (8His-SUMO1-Pba1 with Pba2-FLAG, or 8His-SUMO1-Pba3 with Pba4-HA). Proteins were extracted and incubated with Ni^2+^-NTA resin equilibrated with wash buffer (50 mM Tris-HCl, pH 7.4; 150 mM NaCl, 20 mM imidazole) for 1.5 h at 4 °C with rotation. Afterwards, the resin was washed 3 times with 10 column bed volumes. Bound proteins were eluted in two steps, each with 5 mL elution buffer (50 mM Tris-HCl, pH 7.4; 150 mM NaCl; 15% glycerol; 500 mM imidazole) for 15 min with rotation. Proteins were concentrated using Vivaspin Turbo 4 centrifugal concentrators (10,000 MWCO, Sartorius, Göttingen, Germany), aliquoted, and snap-frozen in liquid nitrogen.

### 2.5. Generation, Purification, and Validation of Anti-α5 and Anti-α6 Antibodies

Antibodies used in this study are listed in Table 3. In this study, α5 and α6 subunits, recombinantly produced in *E. coli* and purified as described in Section 2.4, were used to raise rabbit antisera, with bleedings 60 days after immunization (Pineda Antikörper-Service, Berlin http://www.pineda-abservice.de). To purify antibodies from sera, 250 μg of the *E. coli* purified target protein were loaded on an SDS-PAA gel containing only one well. Proteins were transferred to a PVDF membrane and fixed on the membrane by boiling in water for 30 min. Ponceau S staining was performed for 5 min to visualize the band of the target protein, which was cut and incubated in blocking solution (5% milk-TBST (10 mM Tris, pH 8.0; 150 mM NaCl; 0.05% Tween-20)) for 1 h at room temperature. Afterward, 1 mL of immune serum was diluted 1:25 in 5% milk-TBST and incubated with the membrane at 4 °C overnight. The next day, the membrane was washed in TBST, and antibodies were eluted two times for three minutes using 1 mL glycine elution buffer pH 2.8 (0.1 M glycine, 0.5 M NaCl, 0.05% Tween-20). For neutralization, the antibody solution was transferred to 1 M Tris buffer pH 8.1 containing 10% BSA and 6.5% NaN_3_. Finally, antibodies were stored in aliquots at −80 °C. In order to remove non-specific antibodies binding to unrelated yeast proteins, the following procedure was applied. Yeast strains expressing either α5 or α6 subunits under the control of the galactose-inducible promoter P*_GALS_* (see Table 1) were used. After cells were grown in galactose, they were shifted to a glucose medium overnight to repress the synthesis of the α5 and α6 subunits. Boiled extracts of these strains were loaded onto SDS-PAA gels and proteins were transferred to PVDF membranes. The polyclonal antibody samples were incubated in 5% milk-TBST with the respective membranes to remove material binding nonspecifically to other yeast proteins. For validation, the antibodies were tested by Western blotting against recombinant proteins produced in *E. coli*, as well as against yeast extracts either containing or depleted of the respective α-subunit, or expressing HA-tagged variants displaying slower electrophoretic mobility (Appendix A).

### 2.6. In Vitro Binding Assay on HA-Resin

Equilibrated HA-resin was incubated with 250 μg purified 8His-SUMO1-Pba3-Pba4-HA in a total volume of 1 mL binding buffer (50 mM Tris-HCl, pH 7.4; 150 mM NaCl; 5 mM EDTA; 1% Triton X-100) for 1 h at 4 °C. The resin was then washed 3times with 750 μL binding buffer and briefly vortexed. A total of 50 μg of purified test proteins (8His-SUMO1-α5/-α6/-α7/or -Pba1-Pba2-FLAG) were added to the respective tubes, together with 2 μL SENP1 enzyme to remove the 8His-SUMO1 tag. In a total volume of 1 mL binding buffer, incubation was pursued for 1 h at 4 °C. After washing 3 times with 750 μL binding buffer, samples were briefly vortexed, and proteins were eluted in 100 μL elution buffer (binding buffer + 100 μM HA peptide) for 30 min at 4 °C. Samples were then mixed with Laemmli lysis buffer and analyzed by SDS-PAGE.

### 2.7. In Vitro Assembly of Pba3-Pba4 Complexes

A total of 20 μg of proteins purified from *E. coli* (8His-SUMO1-α5/-α6/-α7/-Pba1-Pba2-FLAG-/or -Pba3-Pba4-HA) were mixed and incubated for 90 min at 4 °C in a total volume adjusted to 100 μL with 26S Buffer. Additionally, samples contained 1 μL SENP1 enzyme to remove the 8His-SUMO1 purification tag. The samples were then mixed with 1xLaemmli lysis buffer (5xLLB: 0.3125 M Tris, pH 6.8; 10% SDS; 50% glycerol; 0.00025% bromophenol blue; 10% β-mercaptoethanol) or 1x native buffer (from 4xNB, see above), and analyzed by SDS-PAGE or native-PAGE, respectively.

## 3. Results

### 3.1. Pba3-Pba4 Is Detected in Several Proteasome Precursor Complexes

We have previously reported a profile of Pba4-containing complexes obtained by gel filtration on a Superdex200 column [5]. To follow Pba3-Pba4, a strain expressing an integrated version of Pba4 tagged with 2xHa in an otherwise wild-type (WT) background was used. Notably, the amount of Pba4 in yeast cells is rather low, such that detection in our hands required highly sensitive ECL reagent and long film exposures of at least 30 min (see Methods, Section 2.2). Pba4-HA eluted in fractions 22–25 (around 350 kDa), and in a second peak that corresponds to complexes with lower molecular weight (MW) in fractions 29–32 (Figure 1A, note that profiles can vary slightly according to the individual gel filtration column used). Complexes containing Pba4 can thus be classified as “low MW” and “high MW”. Complexes comprising Pba4-HA in this otherwise wild-type background are distinct from the 15S-PC (characterized by the presence of Ump1), which is typically confined to fractions 20–22 with the applied chromatography conditions (Appendix A). Further analyses of the fractions obtained from the Superdex200 column by native PAGE revealed additional properties of the separated complexes. Fractions 29-32 contained various low MW complexes with faster mobility in the native gel. In contrast, the higher MW complexes eluted in fractions 22–25 displayed much slower mobility (Figure 1B).

Unfortunately, the high MW complexes tended to dissociate into low molecular weight complexes during various native electrophoresis protocols. The best conditions that allowed us to follow several forms containing Pba4 were BIO-RAD AnykD gels run with Tris/glycine buffer. In other types of native gels, the Pba4-HA signal collapsed into a single band. Even with AnyKD gels, this tendency was observed upon analysis of complexes present in the gel filtration fractions 22–25 (bottom bands in Figure 1B).

Using a strain expressing *PBA4-HA* under transcriptional control of the P*_GAL1_* promoter, genomically integrated at the 3’ UTR of the *URA3* locus in an otherwise wild-type background, a moderate transcriptional expression can be induced for a short period of time by the addition of galactose to the media. After a few minutes of expression, translation of the mRNAs produced from these transgenes can be blocked by the addition of tetracycline, which stabilizes aptamer structures in the 5’ UTR that interfere with translational initiation [17]. As a result, we could follow, over time, the incorporation of HA-tagged Pba4 into nascent precursor complexes by analyzing cell extracts using a combination of native PAGE and anti-HA western blotting (Figure 1C). During the first 30 min of induction, a fast mobility band corresponding to the Pba3-Pba4 dimer was mainly detected. With longer induction times, several bands with slower mobilities appeared, which likely correspond to the high MW forms of Pba3-Pba4 complexes previously detected by gel filtration fractionation.

### 3.2. Pba4-Containing Complexes Are Dependent on α5, α6, and α7 Subunits

Pba3-Pba4 was structurally characterized in a complex containing the α5 subunit [9]. Additionally, this interaction was also observed when α5 is in a complex with α6 and α7 [9,10]. Alignment of the α5-Pba3-Pba4 structure with the α-ring from the mature CP suggests that Pba3-Pba4 is in extensive contact with α4, α5, as well as α6, and less with α7 (Appendix A).

To investigate the contributions of these α-subunits to the formation of the observed high MW complexes, we employed a strategy that enabled termination of the synthesis (shut down, “SD”) of these α-subunits in vivo. To this end, we prepared a set of strains expressing genes encoding the above-mentioned α-subunits from the P*_GALS_* promoter (see methods Section 2.1). In galactose media, the cells express the respective α-subunit-encoding gene and thus behave like WT. After the addition of glucose, transcription of the gene is terminated, resulting in the depletion of the respective subunit. To monitor the effects of the distinct α-subunit SDs on the formation of Pba3-Pba4-containing complexes, the strains also expressed Pba4-HA tagged. Protein extracts from these strains were analyzed by gel filtration and native gel, followed by anti-HA immunoblotting (Figure 2 and Appendix A). Depletion of α5 caused the most dramatic change in the gel filtration profile (Figure 2, upper panel). Specifically, whereas depletion of α5 led to the accumulation of Pba4 complexes mainly in fractions 30 and 31, this form is largely absent from the cells depleted for α6 and α7. The proteins detected in factions 30 and 31 likely correspond to the free Pba3-Pba4 dimer. Instead, the signals were shifted to fractions 29 and 30 when α6 was depleted, and to fractions 28 and 29 when α7 was depleted. The absence of the free Pba3-Pba4 dimer in detectable amounts in the latter strains is likely due to its trapping in early assembly intermediates, preventing its recycling as observed in the WT. The shift by one fraction indicates that Pba3-Pba4 is trapped in a complex with α5 upon depletion of α6, and the additional shift by another fraction caused by the depletion of α7 suggests that in this case, Pba3-Pba4 is trapped in a complex with α5 and α6.

Taken together, the ladder of complexes with fast mobility detected by native PAGE analyses, as shown in Figure 1, can be explained by the following compositions (from low to high MW): free Pba3-Pba4, α5-Pba3-Pba4, α5-α6-Pba3-Pba4, and α5-α6-α7-Pba3-Pba4. These observations suggest that Pba3-Pba4-α5 serves as a starting unit for CP assembly, followed by the incorporation of α6 and α7 subunits.

An identical approach was used to follow the behavior of the Pba4 profile when neighboring subunits in the α- or β-rings were shut down (Appendix A). We did not observe a significant change in the Pba4 gel filtration profile upon depletion of α2, β4, β5, or β6. By contrast, however, α4 depletion changed the profile, resulting in Pba4 detection in additional fractions with higher MW. This was an intriguing result that deserves further consideration. Takagi et al. 2014 [11] found that the Pba3*-*Pba4 heterodimer not only captures α5 but also interacts with α4 through its charged residues. This led them to propose a model in which Pba3*-*Pba4 acts as a molecular matchmaker by increasing the interaction between α5 and α4 while suppressing α4 aggregation, thereby contributing to correct α*-*ring formation. This would be consistent with the idea that α4 is recruited via Pba3-Pba4 as a subsequent step immediately following the formation of the initial α5-α6-α7-Pba3-Pba4 unit. In this view, the higher MW Pba4 complexes detected when α4 is depleted are likely off-pathway complexes formed as a consequence of the failure to incorporate α4. The observation that the formation of these complexes depends on Blm10 might be in line with this interpretation (see Section 3.4).

Another intriguing observation was that individual depletion of either α5, α6, or α7 subunits not only led to the accumulation of distinct low MW complexes in fractions 27–29, as discussed above, but also caused a complete loss of the high MW Pba4-containing complexes detected in fractions 22–25 of wild-type extracts (Figure 2). These findings indicate that α5, α6, and α7 are components of the high MW Pba4-containing complex and are essential for its formation.

### 3.3. Deletion of PBA1 Gene Affects Incorporation of Pba3-Pba4 into High MW Complexes

In mammals, subunits of the PAC1-PAC2 and PAC3-PAC4 chaperones are found simultaneously associated with an α-ring [8]. In yeast, such an α-ring would have a molecular weight of around 200 kDa. If we consider it associated with the corresponding Pba1-Pba2 and Pba3-Pba4 chaperones, we would expect a complex of ~300 kD. However, as mentioned above, we reported that such an α-ring intermediate is not detectable in WT yeast cells [5]. Nevertheless, in order to understand how these two chaperone pairs cooperate in proteasome assembly in yeast, we followed the gel filtration profile of Pba4-HA in a strain deleted for *PBA1* (*pba1∆*). While the previously observed low MW peak was detected, the high MW peak was not detected in the typical fractions 22–25 (Figure 3A, first panel). In the *pba1Δ* strain, the Pba4 peak shifted to fractions 18–22, corresponding to much higher MWs, similar to what was detected upon shut down of α4 (Appendix A). The signals of the low MW Pba4 complexes in fractions 28–31, however, remained unchanged, suggesting that their assembly is independent of the Pba1-Pba2 chaperone. The effect of *PBA1* deletion on Pba4 distribution was also proven by native gel analysis (Figure 3B). We compared crude extracts from a strain overexpressing *PBA4-HA* to strains expressing an integrative *PBA4-HA* in WT, *pba1Δ*, and a strain in which expression of *PBA1* and *PBA2* genes was controlled by the galactose-inducible P*_GAL1_* promoter. In raffinose media, the latter strain behaved similarly to the *pba1∆* strain, whereas in galactose media, the Pba1-Pba2 chaperone was present at levels higher than in WT. As shown in Figure 1, Pba4 was detected in several complexes with distinct fractionation behaviors and mobilities. A slower migrating form accumulated in *pba1∆* and *pba2∆* mutants, or when expression of *PBA1* and *PBA2* was not induced (Figure 3B, raffinose media). Interestingly, a similar slow mobility complex was also observed in *ump1Δ* cells (Appendix A). This is likely due to a depletion of Pba1-Pba2 resulting from a lack of its recycling as a consequence of the maturation impairment caused by *ump1∆* [18]. Strikingly, when Pba1-Pba2 synthesis was induced, this slow-migrating Pba4-containing complex disappeared (Figure 3B, galactose media). As described in Section 3.4, the formation of these complexes depends on Blm10.

### 3.4. Blm10 Deletion Abolishes Pba4-High Molecular Complexes Detected in Absence of PBA1

Pba1-Pba2 associates with the outer surface of the α-ring [18,19,20], where it uses its C-terminal HbYX motifs to insert into the α5-α6 and α6-α7 pockets [21]. The 19S RP ATPases, Blm10, or Cdc48 are other examples of proteins or complexes, binding of which to α-rings involves insertion of their C-termini HbYX motifs into such pockets between α-subunits [22,23,24,25]. All of them have molecular weights large enough to cause a molecular shift in Pba4-HA complexes detectable by our analysis method (gel filtration). The 200 kDa Blm10 protein has been suggested to be involved in proteasome biogenesis [26,27,28] and is found associated either with CPs or precursor complexes (PCs) [29,30]. Its C-terminal residues bind in the pocket between the α5 and α6 subunits, as illustrated in Appendix A [18,22]. Recent studies have shown that, upon overexpression, Blm10 can bind to nascent proteasomal complexes instead of Pba1-Pba2 in vivo [31].

We therefore asked if the absence of Blm10 would alter the Pba4 gel filtration profile by replacing Pba1-Pba2. To answer this question, we prepared a strain expressing Pba4-HA in a *pba1Δ blm10∆* background. In parallel, we generated a *blm10Δ* strain with Pba4-HA. We compared the Pba4 gel filtration profiles of extracts from both strains with the one obtained for WT. The result of these analyses was that the formation of the very high-MW Pba4-HA-containing complex in the absence of Pba1 is dependent on Blm10, as it disappeared in the strain deleted for both genes *PBA1* and *BLM10* (Figure 3A; middle panel). When Pba4-containing complexes were followed by native PAGE analysis, we observed distinct slow mobility complexes: one independent and the other dependent of Blm10 (Figure 3B). In conclusion, Blm10-dependent and Pba4-HA-containing complexes with a very high MW and a corresponding low electrophoretic mobility are formed in the absence of the Pba1-Pba2 chaperone. However, when Pba1-Pba2 is overproduced, as is the case in strains expressing both *PBA1* and *PBA2* from P*_GAL_* promoter in galactose media (Figure 3C), Blm10 presence is insufficient to form such slow mobility Pba4-HA complexes. This observation indicates that Pba1-Pba2 has a higher affinity for these intermediates than Blm10.

These findings suggest that during proteasome biogenesis, Pba3-Pba4-containing complexes at some point transiently associate with the chaperone pair Pba1-Pba2. However, when Pba1-Pba2 amounts are rate-limiting for some reason, Blm10 takes over the available pocket and binds to the Pba3-Pba4-containing complexes.

Because of the observation that Pba1 could be substituted by Blm10 during the biogenesis of Pba4-containing complexes, we asked whether the deletion of both genes would have a synthetic effect on proteasome biogenesis and cell viability of fitness. To our surprise, the double mutant *pba1Δblm10Δ* did not display any apparent growth defect in comparison to the single mutants (Appendix A). The formation of Pba4-containing complexes involving Pba1-Pba2 or Blm10 is not critical for cell viability under the conditions tested. Consistent with this conclusion, we found that similar amounts of 20S CPs containing the α5 subunit were detected in cells lacking functional Pba1-Pba2 and/or Pba3-Pba4 chaperones (Appendix A). Even a yeast mutant lacking all CP assembly chaperones (Ump1, Pba1-Pba2, and Pba3-Pba4) is viable indicating that assembly of functional proteasomes to a live-sustaining extent still occurs in their absence (Appendix A).

### 3.5. Detection of α5-α6-α7-Pba3-Pba4 Complexes in S. cerevisiae

To verify the co-existence of Pba3-Pba4 with the CP subunits α5, α6, and α7 in the same complex, we applied a two-step purification protocol. Crude extracts from WT, pba1*Δ*, or ump1*Δ* strains bearing Pba4 tagged with FLAG-6His (FH) were used for tandem affinity purification of Pba3-Pba4-containing complexes on Ni-NTA and anti-FLAG resins. SDS-PAGE and Western blot analysis revealed that the purified material, in addition to Pba4, contained α5, α6, and α7 (Figure 4), proving that these subunits assemble with the Pba3-Pba4 chaperone, which is consistent with earlier reports [10].

### 3.6. In Vitro Reconstitution of α5-α6-α7-Pba3-Pba4 Precursor Complex Assembly

Earlier in vitro studies demonstrated an α5-dependent interaction between Pba3-Pba4 and the subunits α6 and α7 [10], in line with our observation in yeast cells (Figure 2). In addition, the human PAC1-PAC2, the counterpart of yeast Pba1-Pba2, was shown to bind preferentially to α5 and α7 [7]. According to structural modeling, both chaperone pairs could indeed simultaneously bind to the mentioned subunits. Pba1-Pba2 was found to bind to the outer side of CP α-rings (Appendix A), while Pba3-Pba4 binds to the opposite side of the α5 subunit, which corresponds to the inner side of CP α-rings (Appendix A). We asked if we could reconstitute the assembly of a complex containing the subunits α5, α6 and α7, as well as the chaperone pairs Pba3-Pba4 and Pba1-Pba2 in vitro. To this end, we performed a co-immunoprecipitation experiment as follows: Pba3-Pba4-HA was immobilized on HA resin, and α5, α6, and α7 subunits, as well as Pba1-Pba2-FLAG—all of them carrying N-terminal 8His-SUMO1 tags to increase protein solubility—were added. SENP1, an enzyme that cleaves after SUMO1, was added during the binding step to remove the 8His-SUMO1 tag [32]. After washing, proteins were eluted and analyzed by SDS-PAGE and Western blotting (Appendix A). Detection of α5 and α6 was performed using antibodies generated in this study. Ten percent of the 8His-SUMO1 tagged input was loaded for comparison. All three proteasome subunits (α5, α6, and α7) were efficiently co-eluted with Pba3-Pba4 from the resin. However, Pba1-Pba2 was not detected. Moreover, when the tested proteins were incubated without Pba3-Pba4, no signals were detected, ruling out non-specific binding of these proteins to the resin. The fact that Pba1-Pba2 was not present in the eluted material could be because the affinity to the other proteins is too low to overcome the washing or that Pba1-Pba2 does not engage in the formation of a complex with the other polypeptides. To further analyze whether Pba1-Pba2 could associate with the α5-α6-α7-Pba3-Pba4 complex, we performed an additional in vitro assay. The advantage compared to co-immunoprecipitation on the anti-HA beads, as described above, was the simplicity of the assay, which did not involve any washing steps that could disrupt weak protein–protein interactions. In this assay, the proteins (α5, α6, α7, Pba1-Pba2-FLAG, and Pba3-Pba4-HA) purified from *E. coli* were simply mixed and incubated for 90 min at 4 °C. The N-terminal 8His-SUMO1 tag was cleaved during the assay by SENP1, ensuring proteins stay soluble as long as possible. Samples were analyzed on native polyacrylamide gels and detected using specific antibodies (Figure 5, upper part). The input of each protein was loaded for comparison. Samples were additionally analyzed by SDS-PAGE to prove the presence of the distinct proteins in the particular sample (Figure 5, lower part). After the addition of each polypeptide (Pba3-Pba4 + α5, + α6 + α7, in this order), a staircase-like shift to higher MW complexes was visible, confirming that these proteins indeed assemble into complexes (lanes 6–8). This experiment revealed the order of events taking place. α6 or α7 do not bind to Pba3-Pba4 (lanes 10 and 11), confirming that α5 has to bind first [10]. After binding of α5 to Pba3-Pba4, α6 has to join the complex next, prior to α7. Without α6, α7 is not able to bind to the α5-Pba3-Pba4 complex (lane 12). Upon the addition of Pba1-Pba2, the mobility of the α5-α6-α7-Pba3-Pba4 band did not change in comparison to the sample without Pba1-Pba2 (compare lanes 8 and 9), indicating that Pba1-Pba2 did not bind to the emerging complex.

## 4. Discussion

The specific function of the Pba3-Pba4 chaperone and its cooperation with Pba1-Pba2 in the early steps of proteasome CP assembly in *S. cerevisiae* are poorly understood. The same holds true for their counterparts PAC1-PAC2 and PAC3-PAC4 in mammals. In this study, we provide evidence that Pba3-Pba4 is involved in the formation of an emerging complex containing the proteasome subunits α5, α6, and α7, which represents a starting unit in CP assembly. In vivo experiments involving selective expression shut down of individual α-subunit genes revealed that Pba3-Pba4 assembles consecutively, in this order, with α5, α6, and α7 (Figure 2). Assembly of this starting unit could also be recapitulated in vitro, revealing the same order of individual steps observed in vivo (Figure 5). These experiments showed that the assembly of this unit can occur independently of additional factors.

Yeast Pba3-Pba4 was first described as operating at an early stage of 20S CP assembly, acting before the Ump1 chaperone [33]. Further studies revealed that Pba3–Pba4 binds most strongly to α5, allowing the resolution of an α5-Pba3–Pba4 co-crystal structure [9]. Consistent with our in vitro results, Pba3-6His-Pba4 bacterial lysate mixed with *E. coli* extracts co-expressing α5, α6, and α7 led to the retention of all three α-subunits on Ni^2+^ resin [10]. Interestingly, the lack of synthetic growth defects when *pba3Δ* or *pba4Δ* mutations were combined with *pre9Δ* (lacking the non-essential α3 subunit) suggested that loss of Pba3 or Pba4 proteins, or the absence of the α3 subunit, might affect a common aspect of proteasome assembly or function [10]. We observed that deletion of the *PBA3* gene affects the gel filtration profile of the α4 subunit, reflected by an accumulation of small MW complexes containing this subunit (fractions 25–29) (Appendix A). In addition, native gel analysis showed that the amount of α4 in the 15S-PC was reduced in the *pba3∆* strain compared to WT (Appendix A). This observation aligns with the earlier reported detection of abnormal α-rings lacking α4 in the absence of Pba3-Pba4 [9]. On the other hand, this result was surprising in light of the observation that a lack of Pba3-Pba4 resulted in an accumulation of proteasomes in which a second copy of the α4 subunit replaced α3, correlating with increased resistance to cadmium [10,34]. Based on these findings, Pba3-Pba4 was suggested to be a trans-acting factor whose presence or absence controls the assembly of alternative proteasomal complexes [10]. Our further investigations revealed the accumulation of a distinct α4-containing complex when α2 expression was shut down (SD), the formation of which was not observed when the α3 subunit-encoding *PRE9* gene was deleted or α1 expression was shut down (Appendix A). The α4-containing complex in the α2-SD strain was not detected when any of the other α-subunits was HA-tagged instead of α4. These data indicate that the observed complex is composed only of α4 subunits (Appendix A). Together, the results discussed above, lead us to the conclusion that incorporation of α4 depends on the α2 subunit and is influenced by Pba3-Pba4. Takagi et al. 2014 [11] described the existence of a complex in *pba4Δ*, wherein the ratio of α2 relative to the α6-subunit is twice as high as normal. These authors reported that α5 has an intrinsically higher affinity for α2 than for its own neighbor α4, which led to the proposal that Pba3-Pba4 might contribute to the correct positioning of subunits within the α-rings by suppressing α4 aggregation and enhancing the interaction between α5 and α4 [11].

Thus far, we have not yet developed the in vitro assembly any further by adding either α1 or α4 to the α5-α6-α7-Pba3-Pba4 starting unit. An overlay of the α5-Pba3-Pba4 structure with that of a mature 20S proteasome structure suggests that Pba4 contacts α4 but not α6, which instead is expected to have an extensive contact surface with Pba3 (Figure 6 and Appendix A). Incorporation of the α1 subunit, on the other side, is expected to involve only interactions with α7 and not with Pba3-Pba4. The next subunit to be added to the initial α5-α6-α7-Pba3-Pba4 complex, therefore, is likely α4, which is expected to involve not only interactions with α5 but also with Pba4. Despite this structural model, direct in vitro interactions between Pba3-Pba4 and α3 or α4 subunits, however, were not detected in the absence of α5 [10]. Together, these studies and our results indicate that Pba3-Pba4 not only promotes the formation of the α5-α6-α7-Pba3-Pba4 starting unit, but also controls subsequent correct incorporation of multiple subunits (α2, α3, and α4), which normally occurs after the formation of this starting unit.

A surprising and remarkable result of our work was the inability of the Pba1-Pba2 chaperone to bind to the α5-α6-α7-Pba3-Pba4 starting unit (Figure 5 and Appendix A). Such binding might have been expected based upon structural modeling (Appendix A), but no such binding of Pba1-Pba2 was detectable, at least in vitro. Due to the formation of the α5-α6 and α6-α7 interfaces, which are known to provide pockets for proteins bearing HbYX motifs to bind to nascent CPs, we were expecting to detect binding of Pba1-Pba2 to the emerging α5-α6-α7-Pba3-Pba4 complex. The inability to detect any binding of Pba1-Pba2 in multiple assays and attempts (Figure 5 and Appendix A), however, indicated that α5-α6 and α6-α7 interfaces within the α5-α6-α7-Pba3-Pba4 starting unit are not in a conformation suitable to accommodate the Pba1-Pba2 HbYX motifs. These findings suggest that the binding of Pba1-Pba2 to early proteasome assembly intermediates depends on additional subunits or other factors, the binding of which induces the formation of HbYX binding pockets.

We did not test whether Blm10 or Rpt subunits would bind to the in vitro formed starting unit. However, we found that Blm10 can bind to complexes bearing Pba3-Pba4 in vivo. Blm10, Pba1, and Pba2, as well as several subunits (Rpt2, Rpt3, and Rpt5) of the ATPase module of the 19S regulatory particle, are characterized for having a short C-terminal sequence of amino acids called HbYX motif [2,3,6,35]. The HbYX motifs of Pba1 and Pba2 are required for their ability to bind to the α5-α6 and α6-α7 pockets, respectively (Appendix A) [19,21]. Structurally, Pba1-Pba2 and Blm10 are associated with the outer surface of CP α-rings [18,19,36,37]. The most plausible explanation for the observed high MW Pba4 complex in cells deleted for *PBA1* is that Blm10 substitutes for Pba1-Pba2, which is supported by the absence of these high MW complexes in *pba1∆ blm10∆* double mutants (Figure 6). These observations are in line with the recent description and structural characterization of a 13S-Blm10 complex formed in the absence of Pba1-Pba2 [28]. These and other findings indicate that Blm10 has the capacity to bind to nascent CP intermediates instead of Pba1-Pba2 [26,29,31,38]. Binding of Blm10 instead of Pba1-Pba2 may influence quality control and/or subcellular targeting of nascent proteasomal complexes [31,38,39].

An initial step in CP assembly is binding of α5 to the Pba3-Pba4 chaperone. According to the crystal structure of an in vitro reconstituted α5-Pba3-Pba4 complex [9], Pba3-Pba4 binds to the surface of the α-subunits that will interact with incoming β-subunits in later steps of CP assembly (Figure 6B,C). This explains why the Pba3-Pba4 chaperone dissociates from the complex in later assembly steps involving β-subunits. α5 has extensive contacts with both Pba3 and Pba4, explaining why this α-subunit is the first to establish interactions with the chaperone (Figure 6B) [9]. According to structural modeling, α6 will join the complex by interacting with Pba3 and α5 (Figure 6B). α7 mainly interacts with α6 and has only a small contact surface shared with Pba3 and none with Pba4 (Figure 6B). This model is consistent with the observation that α7 does not join the complex in vitro in the absence of α6 (Figure 5).

## 5. Conclusions

Based upon both our in vivo and in vitro experiments, we propose that early steps of CP assembly involve the formation of a starting unit of Pba3-Pba4 and the proteasome subunits α5, α6, and α7 (Figure 6). Although not yet characterized, a similar starting unit composed of PAC3-PAC4, α5, α6, and α7 is a likely intermediate in mammalian cells. At least in yeast, the formation of this unit occurs independently of Pba1-Pba2, which we assume to act at later assembly steps involving other α-subunits.

## Figures and Tables

**Figure 1 biomolecules-15-00683-f001:**
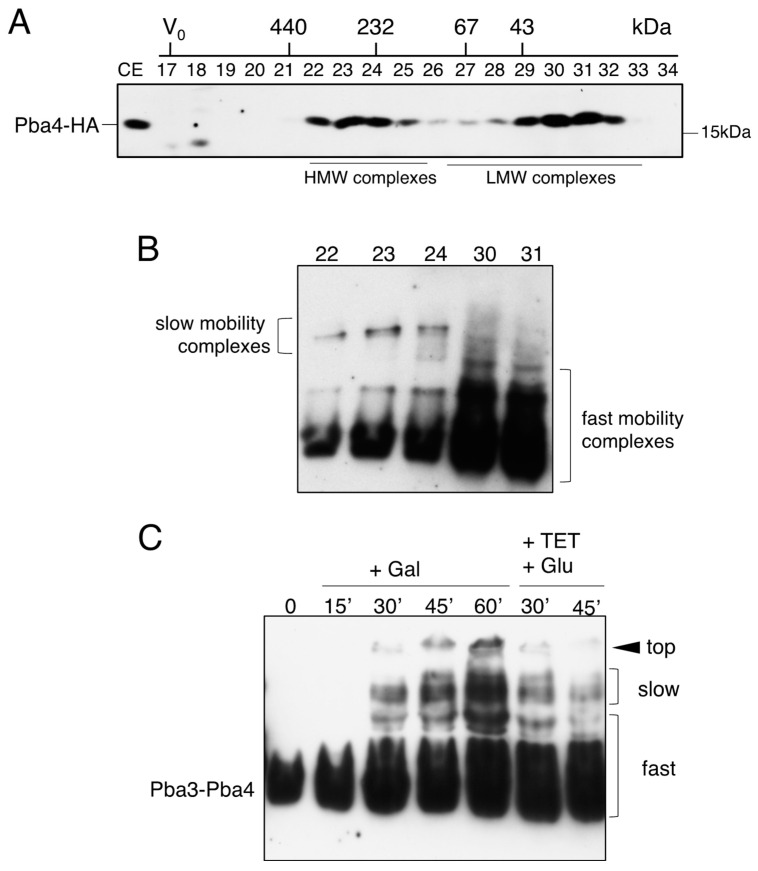
Characterization of complexes containing Pba3-Pba4 in yeast. (**A**) Superdex200 profile of cell extracts: Pba4-HA is detected in low molecular weight (LMW) complexes and a ca. 350 kD high molecular weight (HMW) complex. (**B**) Analysis of the indicated Superdex200 fractions by native gel followed by immunoblot detection of Pba4-HA. The low and fast mobility forms are indicated. (**C**) Expression of *PBA4* is controlled at the level of transcription by P*_GAL1_* and at the level of translation by the tc3 aptamer. Cells were grown in a medium containing raffinose. Galactose was added to the medium at time point 0, followed by the indicated incubation times. After 60 min, expression was stopped by the addition of glucose and tetracycline. Cells were collected at the indicated time points, and the extracts were subjected to a native gel analysis. Original images can be found in Appendix A.

**Figure 2 biomolecules-15-00683-f002:**
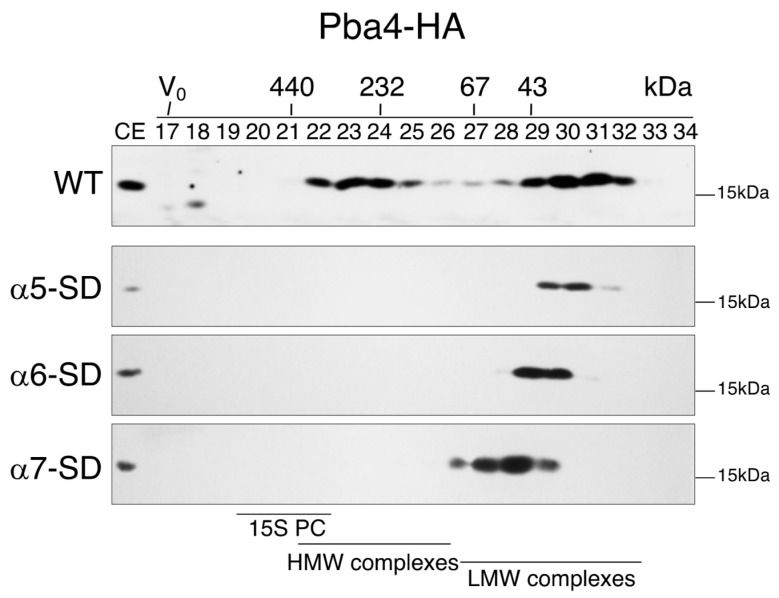
Formation of Pba3-Pba4 complexes depends on the consecutive assembly of subunits α5, α6, and α7. Superdex gel filtration profiles detecting Pba4-HA in extracts from strains wherein P_GALS_-controlled expression of the indicated α-subunits was individually shut down (SD) by the addition of glucose. Proteins in the indicated fractions were separated by SDS-PAGE and subjected to anti-HA immunoblotting. Note that the first panel for the wild type (WT) is identical to the one shown in Figure 1A for better comparison. CE, whole cell extract. Positions of molecular weight markers are indicated. Original images can be found in Appendix A.

**Figure 3 biomolecules-15-00683-f003:**
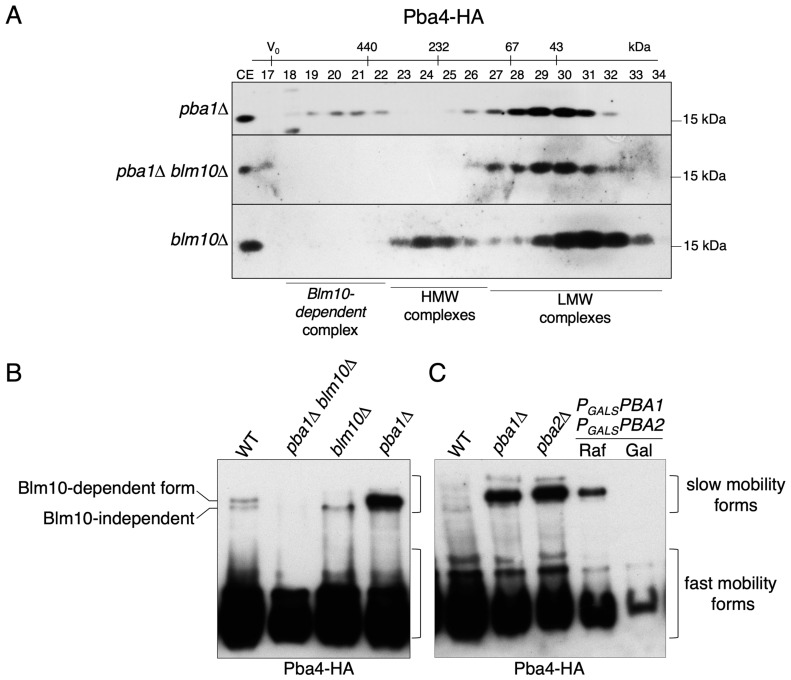
Deletion of the *PBA1* gene causes the accumulation of Pba4 complexes containing Blm10. (**A**) Superdex200 gel filtration analysis: Blm10 deletion abolishes high MW complexes containing Pba4 detected in the absence of Pba1. CE, whole cell extract. (**B**) Native PAGE analysis with western blot detection of Pba4-HA was performed with the indicated strains. All strains expressed genomic versions of PBA4-HA and carried either deletions of the indicated genes or contained both PBA1 and PBA2 genes under the control of the galactose-inducible P*_GALS_*promoter. (**C**) Same as in (**B**), but with additional strains as indicated. Original images can be found in Appendix A.

**Figure 4 biomolecules-15-00683-f004:**
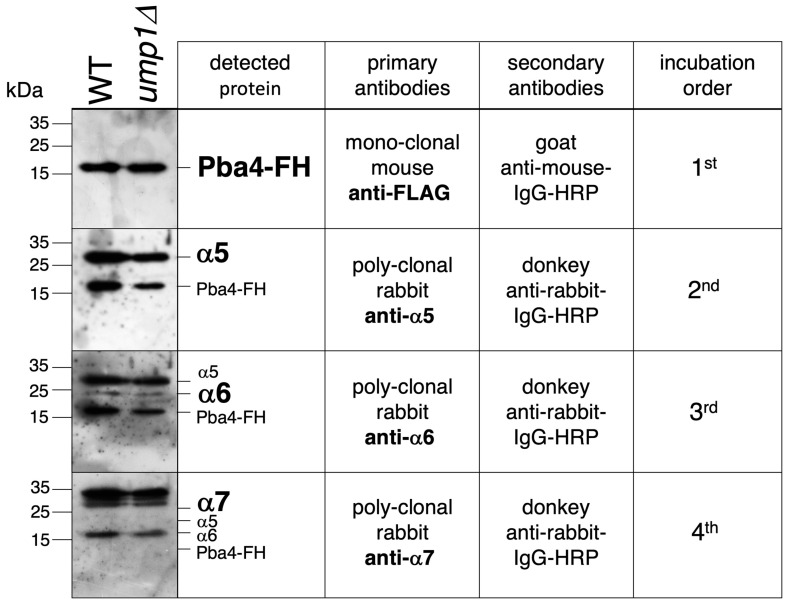
Analysis of Pba4-FH-containing complexes purified from cells from 2 L cultures of wild-type (WT) and ump1*Δ* cells by SDS-PAGE and immunoblotting. Tested was the presence of FLAG, *α*5, *α*6, and *α*7 signals. The order of antibody incubation, using the same membrane without in-between stripping, was as indicated. Original images can be found in Appendix A.

**Figure 5 biomolecules-15-00683-f005:**
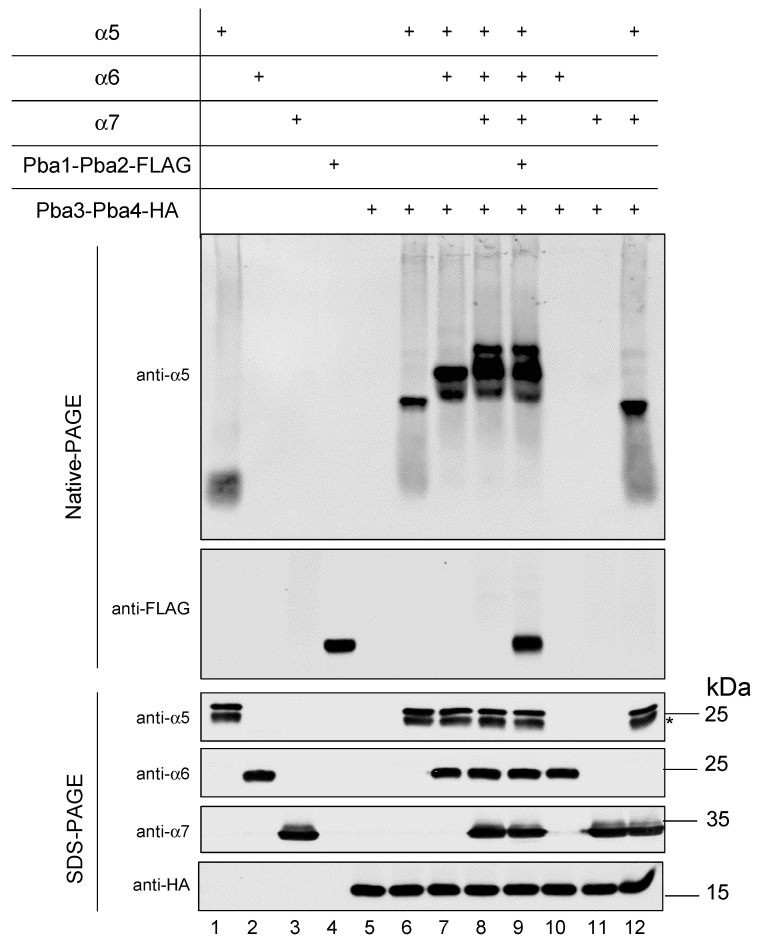
Native PAGE analysis of in vitro assembled complexes involving Pba3-Pba4. Proteins were expressed in *E. coli* and purified using their N-terminal 6His-SUMO1 tag. Purified α-subunits and/or Pba1-Pba2 were sequentially added to Pba3-Pba4 (in the order indicated above, from top to bottom) and incubated at 4 °C for 90 min in the presence of SENP1 to remove 6His-SUMO1 tags. The formation of complexes in the different samples was analyzed by native gel electrophoreses and immunoblotting with anti-α5 and anti-FLAG antibodies (upper part). The presence of the distinct proteins in all the assays was confirmed by SDS-PAGE and western blotting using the indicated antibodies (bottom part). An asterisk indicates the position of an α5 degradation product. Original images can be found in Appendix A.

**Figure 6 biomolecules-15-00683-f006:**
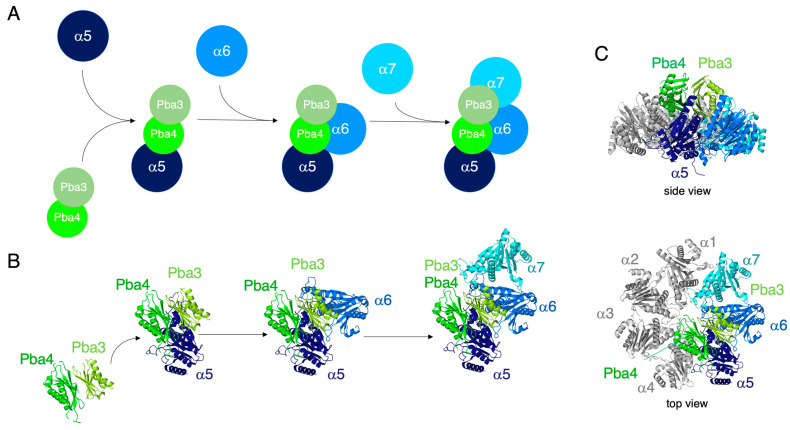
Model of the role of chaperone Pba3-Pba4 in early steps of proteasome assembly. (**A**) Schematic representation of the order of steps in the formation of the α5-α6-α7-Pba3-Pba4 starting unit. Pba3-Pba4, both in vivo and in vitro, first associates with the α5 subunit. This unit subsequently associates with α6, followed by α7. (**B**) Structural modeling of the early intermediates formed by Pba3-Pba4, based upon the crystal structure of the α5-Pba3-Pba4 complex (PDB: 2Z5C [9]) and neighboring α-subunits based upon the structure of the late-PC (PDB: 8RVL [18]). (**C**) Modeling of Pba3-Pba4 onto the inner surface of an α-ring structure taken from the late-PC (shown in side and top views).

**Table 1 biomolecules-15-00683-t001:** Strains used in this study.

Name	Genotype	Source
BY4741	*MAT*a *his3Δ1 leu2Δ0 metΔ0 ura3Δ0*	EuroScarf
CM2	*MAT*a *PRE6-HA::YIplac211*	[5]
CM134	*MAT*a *P_GALS_-SCL1* (α1)::*kanMX PRE6* (α4)-*HA::YIplac211*	[5]
CM144	*MAT*a P*_GALS_-PRE6* (α4)::*KanMX PRE6* (α4)-*HA::YIplac211*	this study
CM192	*MAT*a *pba1Δ::HIS3 PBA4-HA::YIplac211*	this study
CM223	*MAT*a *pre9Δ::HIS3 PBA4-HA*::*YIplac211*	this study
CM233	*MAT*a P*_GALS_-PRE10* (α7)::*KanMXPBA4-HA*::*YIplac211*	this study
CM241	*MAT*a P*_GALS_-PRE6* (α4)::*KanMXPBA4-HA*::*YIplac211*	this study
CM257	*MAT*a *PRE9* (α3)-*HA::YIplac211*	this study
CM310	*MAT*a *pba1Δ::NAT pba3Δ::HIS3 PUP2* (α5)-*HA::YIplac211*	this study
CM342	*MAT*a *P_GAL1_-PUP1* (β2)::*KanMX PBA4-HA::YIplac211*	this study
FP8	*MAT*a *PRE6* (α4)-*HA::YIplac211 pre9Δ::HIS3MX6*	[5]
FP16	*MAT*a *PBA4-HA*::*YIplac211*	[5]
JD47-13C	*MAT*a *his3*-*Δ200 leu2-3,112lys2-801 trp1*-*Δ63 ura3-52*	[13]
JD129	*MAT*a *UMP1-HA::YIplac128*	[13]
MN6	*MAT*a *ump1Δ::HIS3MX PBA4-HA::YIplac211*	this study
MN48	*MAT*a P*_GAL1_-PRE1* (β4)::*KanMXPBA4-HA*::*YIplac211*	this study
MN49	*MAT*a P*_GAL1_-PRE2* (β5)::*KanMXPBA4-HA*::*YIplac211*	this study
MN50	*MAT*a P*_GAL1_-PRE7* (β6)::*KanMXPBA4-HA*::*YIplac211*	this study
MN55	*MAT*a *blm10Δ::KanMX PBA4-HA::YIplac211 (BY4741)*	this study
MO47	*MAT*a *P_GAL1_-PBA1::TRP1 P_GAL1_ PBA2::HIS3* *PBA4-HA::YIplac211*	this study
PR20	*MAT*a P*_GALS_-SCL1* (α1)::*KanMXPBA4-HA*::*YIplac211*	this study
PR21	*MAT*a P*_GALS_-PRE8* (α2)::*KanMXSCL1* (α1)-*HA::YIplac211*	this study
PR22	*MAT*a P*_GALS_-PRE8* (α2)::*KanMXPRE8* (α2)-*HA::YIplac128*	this study
PR23	*MAT*a P*_GALS_-PRE8* (α2)::*KanMXPRE9* (α3*)-HA::YIplac211*	this study
PR24	*MAT*a P*_GALS_-PRE8* (α2)::*KanMXPRE6* (α4*)-HA::YIplac211*	this study
PR25	*MAT*a P*_GALS_-PRE8* (α2)::*KanMXPUP2* (α5)-*HA::YIplac211*	this study
PR26	*MAT*a P*_GALS_-PRE8* (α2)::*KanMXPRE5*(α6)-*HA::YIplac211*	this study
PR27	*MAT*a P*_GALS_-PRE8* (α2)::*KanMXPRE10* (α7)-*HA::YIplac211*	this study
PR28	*MAT*a P*_GALS_-PRE8* (α2)::*KanMXUMP1-HA::YIplac128*	this study
PR30	*MAT*a P*_GALS_-PRE8* (α2)::*KanMXPBA4-HA*::*YIplac211*	this study
PR60	*MAT*a P*_GALS_-PUP2*(α5)::*KanMXPBA4-HA*::*YIplac211*	this study
PR70	*MAT*a P*_GALS_-PRE5*(α6)::*KanMXPBA4-HA*::*YIplac211*	this study
PR118	*MAT*a *pba1Δ::HIS3*	this study
PR119	*MAT*a *pba3Δ::HIS3*	this study
PR130	*MAT*a *pba1Δ::HIS3 PUP2* (α5)-*HA::YIplac211*	this study
PR132	*MAT*a *pba3Δ::HIS3 PRE6* (α4)-*HA::YIplac211*	this study
PR193	*MAT*a *pba3Δ::HIS3 SCL1* (α1)-*HA::YIplac211*	this study
PR194	*MAT*a *pba3Δ::HIS3 PRE8* (α2)-*HA::YIplac211*	this study
PR207	*MAT*a *pba1Δ::HIS3 blm10Δ::KanMX*	this study
PR211	*MAT*a*PBA4-FLAG6His* ::*YIplac204*	this study
PR213	*MAT*a*blm10Δ::KanMX pba1Δ::HIS3* *PBA4-HA::YIplac211 (BY4741)*	this study
PR236	*MAT*alpha *pba1Δ::NAT1*	this study
PR238	*MAT*alpha *pba1Δ::NAT1 pba3Δ::HIS3*	this study
PR241	*MAT*alpha *ump1Δ::LEU2 pba3Δ::HIS3*	this study
PR244	*MAT*a *pba1Δ::NAT1 pba3Δ::HIS3*	this study
PR245	*MAT*a *ump1Δ::LEU2*	this study
PR247	*MAT*a *pba1Δ::NAT1 ump1Δ::LEU2 pba3Δ::HIS3*	this study
PR265	*MAT*a *pba2**Δ::KanMX PBA4-HA*::*YIplac211**(BY4741)*	this study
PR344	*MAT*a*ump1Δ::HIS3 PBA4-FLAG-6His*::*YIplac204*	this study
PR357	*JD47-13C P_GAL1_-tc3-PBA4-2HA::URA*	this study
PR368	*MAT*a*pba1Δ::TRP1 ump1Δ::HIS3 PBA4-HA::YIplac211*	this study
*blm10Δ*	*MAT*a *blm10Δ:: KanMX (BY4741)*	EuroScarf
*pba1Δ*	*MAT*a *pba1Δ:: KanMX (BY4741)*	EuroScarf

**Table 2 biomolecules-15-00683-t002:** Plasmids used in this study.

Name	Design	Source
pJD791	pUC21-P*_GAL1_*-tc3-*PBA4*-2HA	this study
pJZ20	pET11a-*8His-SUMO1-PUP2(α5)*	this study
pJZ21	pET11a-*8His-SUMO1-PRE5(α6)*	this study
pJZ22	pET11a-*8His-SUMO1-PRE10(α7)*	this study
pJZ24	pET11a-*8His-SUMO1-PBA1-PBA2-FLAG*	this study
pJZ39	pET11a-*8His-SUMO1-PBA3-PBA4-HA*	this study

**Table 3 biomolecules-15-00683-t003:** Antibodies used in this study.

Antibody	Host	Dilution	Source
anti-HA (3F10)	rat	1:2000	Roche
anti-HA (3F10)-peroxidase	rat	1:2000	Roche
anti-FLAG (M2)	mouse	1:1000	Sigma-Aldrich
anti-Pre10/a7	rabbit	1:1000	[12]
anti-Pup2/a5	rabbit	1:1000	this study
anti-Pre5/a6	rabbit	1:1000	this study
anti-mouse-IgG-HRP	goat	1:5000	Sigma-Aldrich
anti-rabbit-IgG-HRP	donkey	1:5000	GE healthcare
anti-mouse Alexa Fluor Plus 680	goat	1:5000	Thermo Fisher
anti-rabbit Alexa Fluor Plus 800	goat	1:5000	Thermo Fisher
anti-rat Alexa Fluor 680	goat	1.5000	Thermo Fisher

## Data Availability

Data are contained within the article or Appendix A.

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
