# Peer review of "The α5-α6-α7-Pba3-Pba4 Complex: A Starting Unit in Proteasome Core Particle Assembly"

_biomolecules, 2025, doi:10.3390/biom15050683_

Round 1

Reviewer 1 Report

Comments and Suggestions for Authors

In this study, the authors characterized Pba3-Pba4 chaperone complex  and found that, in Saccharomyces cerevisiae cells, the chaperone dimer associates with subunits α5, α6, and α7 of the proteasome 20S core particle (CP) . They demonstrated the formation of the Pba3-Pba4 α5 followed by α6, and α7  into the nascent CP complex in in vitro assays, providing insight into the molecular sequence of events leading to the assembly of proteasome CP. Furthermore, in the absence of functional Pba1-Pba2 or Ump1, the authors detected Pba4 in complexes containing Blm10, suggesting an alternative assembly pathway. Overall, this is a well-planned and executed study that will be of interest to researchers studying the  mechanisms regulating proteasome assembly.

As it stands, I did not find anything particularly concerning or requiring major revision. The study is well-planned and sheds light on a relatively complex aspect of proteasome structure, specifically, how the 20S CP is assembled with the assistance of chaperones. It clearly fills a significant gap in the field. In this study, the authors demonstrate that the Pba3-Pba4 complex is responsible for mediating the association of the α5  α6 and α7 subunits in a sequential manner. The conclusions drawn by the authors is supported by the results in the manuscript which show the association of Pba3-Pba4 with proteasome  α subunits by fractionation, immunoblotting and in vitro assays using recombinant proteins . The data presented is generally robust, although some of the immunoblots are of suboptimal quality. That said, considering the experimental setup (fractionation and immunodetection in yeast strains), these experiments can be technically challenging, and I would not be overly critical of this aspect. One suggestion for improvement would be to incorporate some of the structural modeling from Supplementary Figure S2 into the main text. As it stands, Figure 6 in the main manuscript is somewhat underwhelming. Including the structural model illustrating the association of Pba3-Pba4 with the core particle as part of Figure 6 would strengthen the conclusions and improve the overall impact of the paper.

Author Response

In this study, the authors characterized Pba3-Pba4 chaperone complex  and found that, in Saccharomyces cerevisiae cells, the chaperone dimer associates with subunits α5, α6, and α7 of the proteasome 20S core particle (CP) . They demonstrated the formation of the Pba3-Pba4 α5 followed by α6, and α7 into the nascent CP complex in in vitro assays, providing insight into the molecular sequence of events leading to the assembly of proteasome CP. Furthermore, in the absence of functional Pba1-Pba2 or Ump1, the authors detected Pba4 in complexes containing Blm10, suggesting an alternative assembly pathway. Overall, this is a well-planned and executed study that will be of interest to researchers studying the mechanisms regulating proteasome assembly.

We thank the reviewer for the positive feedback on our study acknowledging that our study is well-executed and will be of interest to researchers studying proteasome assembly.

As it stands, I did not find anything particularly concerning or requiring major revision. The study is well-planned and sheds light on a relatively complex aspect of proteasome structure, specifically, how the 20S CP is assembled with the assistance of chaperones. It clearly fills a significant gap in the field. In this study, the authors demonstrate that the Pba3-Pba4 complex is responsible for mediating the association of the α5  α6 and α7 subunits in a sequential manner. The conclusions drawn by the authors is supported by the results in the manuscript which show the association of Pba3-Pba4 with proteasome  α subunits by fractionation, immunoblotting and in vitro assays using recombinant proteins . The data presented is generally robust, although some of the immunoblots are of suboptimal quality. That said, considering the experimental setup (fractionation and immunodetection in yeast strains), these experiments can be technically challenging, and I would not be overly critical of this aspect. One suggestion for improvement would be to incorporate some of the structural modeling from Supplementary Figure S2 into the main text. As it stands, Figure 6 in the main manuscript is somewhat underwhelming. Including the structural model illustrating the association of Pba3-Pba4 with the core particle as part of Figure 6 would strengthen the conclusions and improve the overall impact of the paper.

We have exchanged a couple of western blots (Figure 1B and 3C, the latter is 3B in the revised manuscript) by better versions to improve quality (see also response to reviewer 2)

We appreciate the suggestion to sustain our cartoon model in figure 6 with structural modeling. To do so, we illustrate now in the new figure 6B how Pba3-Pba4 interacts with a-subunits during the assembly of the a5-a6-a7-Pba3-Pba4 starting unit based upon the known crystal structure of a5-Pba3-Pba4 (reference [9]) and the cryo-EM structure of a late CP assembly intermediate bearing a fully assembled a-ring (reference [24]).

Reviewer 2 Report

Comments and Suggestions for Authors

The manuscript by Matias et al. “The a5-a6-a7-Pba3-Pba4 complex, a starting unit in proteasome core particle assembly” describes an interesting issue – the first steps of proteasome assembly in yeasts. The scientific significance of the study is linked to the principle role played by proteasomes in cells. These complexes degrade most intracellular proteins and hence are involved in almost all aspects of the cellular metabolism. Therefore, elucidation of the sequence of events during proteasome formation is an important and interesting task. Though, a lot has been done on the issue, several points need to be clarified. Matias et al. performed quite a lot of experiments (I was impressed by the amount of yeast strains that were used and generated), aimed to reveal the initiation of proteasome alpha-ring assembly, involvement of proteasome assembly chaperones and a proteasome regulator PA200. Authors obtained interesting results, however I have several points that need to be addressed before the paper can be published.

Major points:

  • Authors mention acquisition of antibodies anti-α5, α6, α7 in the study. However, I could hardly find detailed information regarding how it was done. What animals were used, what and how peptides were selected for the immunization, how animals were kept, were they treated in accordance with the ethical norms, how many bleedings were performed and the antibody purification procedure itself is misty. Importantly, the text lacks data on validation of these antibodies before use. Since utilization of the antibodies provides important results, I assume that a specific part in materials and methods should be added, as well as the figure/s in Supplementary material, despite authors demonstrate reactivity of the obtained antibodies with the recombinant proteins in a “pure” system. In addition, to my point of view recombinant alpha subunits also need to be validated following purification with commercial antibodies or MS.

  • Figure 1C. Authors indicate that Pba3-Pba4 complex is detected, however, only the antibodies to HA-tagged Pba4 were used, thus, I assume that presence of Pba3 also needs to be confirmed by Wb. Figure 2 and text Lanes 235-238. Authors indicate that: “Pba3-Pba4 is trapped in the complex with α5 upon depletion of α5, and the additional shift by another fraction caused by the depletion of α7 suggests that in this case Pba3-Pba4 is trapped in a complex with α5 and α6”. It is likely to be so, but again, I think that confirmation of that fact by Wb with antibodies to α5, α6, α7 is desirable, considering that specific antibodies were obtained by the authors.

  • I was surprised by the role of the Blm10, which is a nuclear proteasome activator, within the initial steps of alpha ring assembly. Do authors suppose it can somehow stimulate assembling proteasome import into the nucleus, or it has a chaperone function? Though authors used mutant strains with deletion of Blm10 as controls, I think, presence of Blm10 in high molecular weight complex should be confirmed by Wb with specific antibodies.

  • Do fractions 22-25 and slow mobility complexes contain Pba1-Pba2? Recombinant proteins purified from bacteria might have alterations in confirmation, thus their ability to form a correct complex with chaperones might be altered hampering for instance association with Pba1-Pba2 in the in vitro Please comment on that. Do Pba1-Pba2 and Blm10 use the same alpha pockets?

  • ΔPba1 is not shown in Figure4. Moreover, were multiply strippings and reprobings performed with a single blot? If so it might significantly affect the quality of the results especially during late reprobings.
  • Figure 1B. It is quite difficult to make conclusions from the presented blot. I understand the difficulties, but would it be possible to provide a better image?
  • Figure 3B. Should we expect still to see the Pba4-HA positive slow mobility forms (fractions 22-25) in WT in Gal media? In addition there is a huge discrepancy between strength of the slow mobility fraction signal in pba1Δ samples in the right (hardly visible signal) and in the left blot (strong signal).
  • Figure 5. Please explain double band for alpha5.
  • Lane 384. Please comment on the incubation of protein mixture at 40C for the in vitro reconstitution of the complex.
  • Perhaps few phrases should be added into the introduction indicating what are proteasomes and their importance for the cells.

Minor points:

  • Lane 204 contain extra text;
  • Lane 228 I could not find Figure S3C and D in Supplementary material;
  • Western blots in general, lack positions of molecular weight markers;
  • What is a CE Figure1,2,3, please indicate;
  • Please indicate what is HMW and LMW in Figure legends;
  • The text has some typos.

Author Response

The manuscript by Matias et al. “The a5-a6-a7-Pba3-Pba4 complex, a starting unit in proteasome core particle assembly” describes an interesting issue – the first steps of proteasome assembly in yeasts. The scientific significance of the study is linked to the principle role played by proteasomes in cells. These complexes degrade most intracellular proteins and hence are involved in almost all aspects of the cellular metabolism. Therefore, elucidation of the sequence of events during proteasome formation is an important and interesting task. Though, a lot has been done on the issue, several points need to be clarified. Matias et al. performed quite a lot of experiments (I was impressed by the amount of yeast strains that were used and generated), aimed to reveal the initiation of proteasome alpha-ring assembly, involvement of proteasome assembly chaperones and a proteasome regulator PA200. Authors obtained interesting results, however I have several points that need to be addressed before the paper can be published.

We thank the reviewer for the feedback and constructive criticism and for recognizing that we have obtained interesting results in an important and interesting task, the elucidation of the sequence of events during proteasome formation.

Major points:

  • Authors mention acquisition of antibodies anti-α5, α6, α7 in the study. However, I could hardly find detailed information regarding how it was done. What animals were used, what and how peptides were selected for the immunization, how animals were kept, were they treated in accordance with the ethical norms, how many bleedings were performed and the antibody purification procedure itself is misty. Importantly, the text lacks data on validation of these antibodies before use. Since utilization of the antibodies provides important results, I assume that a specific part in materials and methods should be added, as well as the figure/s in Supplementary material, despite authors demonstrate reactivity of the obtained antibodies with the recombinant proteins in a “pure” system. In addition, to my point of view recombinant alpha subunits also need to be validated following purification with commercial antibodies or MS.

We thank the reviewer for this suggestion. We have now added a section to the Materials and Methods section, in which we describe in detail how the proteins for immunization were obtained and how the antibodies were raised in rabbits, which was performed by a company (Pineda Antikörper-Service Berlin; http://www.pineda-abservice.de) work of which, to the best of our knowledge, adheres to ethical norms and is scrutinized by the authorities. We have added the new Supplementary figures S1 to illustrate the approach of antigen production and the validation of antibody specificity. Recombinant alpha subunits which were produced in E. coli as fusions to 8xHis-SUMO1 from sequenced plasmids and purified by Metal affinity chromatography with highly specific elution with SUMO-specific protease SENP1. Their validation was an indirect result of the antibody validation. Specifically, the antibodies raised were shown to recognize both the purified recombinant protein as well as the endogenous protein in yeast extracts. The specificity of the latter recognition was verified by comparing extracts from yeast cells expressing size-shifted tagged variants of a given alpha-subunit to extracts from cells with the untagged authentic version. Antibodies raised against α7 have been introduced and validated earlier (Ramos et al., 2004; reference [12]).

  • Figure 1C. Authors indicate that Pba3-Pba4 complex is detected, however, only the antibodies to HA-tagged Pba4 were used, thus, I assume that presence of Pba3 also needs to be confirmed by Wb. Figure 2 and text Lanes 235-238. Authors indicate that: “Pba3-Pba4 is trapped in the complex with α5 upon depletion of α5 (probably should be and the additional shift by another fraction caused by the depletion of α7 suggests that in this case Pba3-Pba4 is trapped in a complex with α5 and α6”. It is likely to be so, but again, I think that confirmation of that fact by Wb with antibodies to α5, α6, α7 is desirable, considering that specific antibodies were obtained by the authors.

Pba3 and Pba4 were shown by other to act as heterodimer. Genetic and biochemical data had shown that one of the subunits does not have a function without the other, and ends up insoluble without the other (Le Tallec et al., 2007; Kusmierczyk et al., 2008; references [37| and [10], respectively). We do not have a sensitive antibody against Pba4. Furthermore, our antibodies are not sensitive enough to detect α5 or α6 fractions from gel filtration because fractionation of crude cell extract goes along with a further dilution of the complexes to an extend that has precluded a reasonable detection of these subunits in our hands. We therefore decided to rely on the highly sensitive detection of 2xHA tagged Pba4 for identification of the complexes of interest. Our point was to show how the depletion of individual α-subunits affects such complexes. As acknowledged by the reviewer, our evidence that shutting down expression of individual α-subunits each causes a distinct and specific shift in the fractionation behavior of Pba4-HA containing complexes towards lower molecular weights is very likely due to an accumulation of incomplete Pba3-Pba4-HA starting units lacking certain subunits (Pba3-Pba4-α5-α6 in the case of α7 depletion, and Pba3-Pba-α5 in the case of α6 depletion).

I was surprised by the role of the Blm10, which is a nuclear proteasome activator, within the initial steps of alpha ring assembly. Do authors suppose it can somehow stimulate assembling proteasome import into the nucleus, or it has a chaperone function? Though authors used mutant strains with deletion of Blm10 as controls, I think, presence of Blm10 in high molecular weight complex should be confirmed by Wb with specific antibodies.

Blm10 has not been a focus of this study, which was on the role of Pba chaperones. However, it was an observation made during our experiments with strains lacking or depleted for Pba1 that Blm10 binds in higher amounts if Pba1-Pba2 is absent. As indicated in the discussion, we therefore did not follow up further on this observation, and hence refrained from speculating about the function of Blm10 during CP biogenesis. In the discussion, instead, we pointed to other studies linking Blm10 to quality control and subcellular targeting of proteasomal complexes (“Blm10 binding instead of Pba1-Pba2 might have impacts on quality control and/or subcellular targeting of nascent proteasomal  complexes [35,42,43]”).

  • Do fractions 22-25 and slow mobility complexes contain Pba1-Pba2? Recombinant proteins purified from bacteria might have alterations in confirmation, thus their ability to form a correct complex with chaperones might be altered hampering for instance association with Pba1-Pba2 in the in vitro Please comment on that. Do Pba1-Pba2 and Blm10 use the same alpha pockets?

Fractions 22-25 (in figures 1A, 2, or 3) were all obtained by fractionation of yeast cell crude extracts. (No recombinant proteins purified from bacteria were involved in these experiments. Such recombinant proteins were only employed in the experiments shown in figures 5 and S6.) The exact nature of complexes, detected in fractions 22-25, is unknown. However, it is clear that the signals observed for the wild-type in these fractions move to fractions 18-22 in a strain lacking Pba1 (see figure 3), indicating that formation of complexes in fraction 22-25 depends on Pba1-Pba2. The abundance of these intermediates, however, is very low and therefore only detectable upon detection of the 2xHA tagged Pba4 after long exposures with super sensitive ECL reagents (Femto). Our attempts to purify and further characterize this complex by MS have failed up until now.

  • ΔPba1 is not shown in Figure4. Moreover, were multiply strippings and reprobings performed with a single blot? If so it might significantly affect the quality of the results especially during late reprobings.

A purification of Pba4-FLAG-6xHis complexes was not performed for pba1∆ because our intention was to verify the conclusion derived from the α -subunit shutdown experiments (figure 2) that Pba3-Pba4 are together in complexes with α5, α6, and α7, which we detect in fractions 22-25 as discussed above. As complexes with this fractionation behavior disappear in the absence of Pba1 (Figure 3), we did not attempt to purify them from this background. As discussed in another section, the loss of Pba3-Pba4 complexes of this size is apparently due to binding of Blm10 instead of Pba1-Pba2.

We agree with the reviewer that the probing regiment was not explained sufficiently. We had tried to express in the figure legend that the same blot was probed multiple times (“The detection order of antibody incubation was as indicated”). Importantly, the blots were not stripped, but instead simply reprobed to avoid losing any signals due to the stripping protocol. This explains why the signals of Pba4-HA is also detectable in the subsequent reprobings with anti-α-subunit antibodies, as is α5 in subsequent reprobings with antibodies against the other α-subunits. Only the week signals for α6 are no longer detectable after reprobing with α7 apparently due to the additional washing steps. We have revised the labeling within the figure as well as the figure legend to make this clear.

  • Figure 1B. It is quite difficult to make conclusions from the presented blot. I understand the difficulties, but would it be possible to provide a better image?

Indeed, the native gel pattern of these intermediates is complex. Furthermore, as we have acknowledged, the high molecular weight (HMW) complexes have a tendency to dissociate in native gels. What this image was intended to show is that complexes in fractions 22-25 have a mobility distinct from those present in gel filtration fractions 29-32. In the revised manuscript, we have replaced figure 1B by a better version obtained in an independent replicate.

  • Figure 3B. Should we expect still to see the Pba4-HA positive slow mobility forms (fractions 22-25) in WT in Gal media? In addition there is a huge discrepancy between strength of the slow mobility fraction signal in pba1Δ samples in the right (hardly visible signal) and in the left blot (strong signal).

The native PAGE analysis shown in figure 3B complements data from gel filtration analyses presented in figure 3A, which indicated that the low mobility Pba4-containing complexes depend on Blm10. Pba1-Pba2 and Blm10 binding is mutually exclusive because they both require binding via C-terminal HbYX motifs to the α5-α6 pocket (shown in Supplementary figure S4). Higher amounts of Blm10-dependent Pba4-HA-containing complexes are therefore found in strains lacking or depleted for Pba1-Pba2. Conversely, when Pba1-Pba2 is overproduced, as is the case in the strain expressing both PBA1 and PBA2 from GAL promoter in galactose media, Blm10 presence is insufficient to form slow mobility Pba4-HA complexes. The answer to the reviewer’s question therefore is: “No, we do expect slow mobility Pba4-HA forms caused by Blm10 association”. These data suggest that Pba1-Pba2 binds to nascent CP intermediates even in the presence of Blm10. The latter may act as an alternative to Pba1-Pba2 under conditions that lead to perturbations in biogenesis such as mutations that slow down assembly. We have revised the main text to further explain this conclusion. This finding is in line with the observation by Kaur et al. (2024; references [32]) that Pba1-Pba2 can bind alternatively to nascent α-ring surfaces.

We thank the reviewer for pointing to this seeming discrepancy which is due to some variation from preparation to preparation. We have replaced figure 3C by a more representative blot from a replicate experiments with similar strength of bands for the complex in the pba1∆ strain. To have a better fit with the revised version of the main text, figures 3B and 3C have been flipped in the revised version as compared to the original manuscript.

  • Figure 5. Please explain double band for alpha5.

The faster migrating band appears to be a degradation product already in the input material of the purified α-subunit. We have indicated this in the revised figure and legend.

  • Lane 384. Please comment on the incubation of protein mixture at 40C for the in vitro reconstitution of the complex.

We chose 4°C in order to avoid loss of function of proteins due to unfolding or non-specific aggregation occurring in the absence of any general folding chaperones that we have previously found to influence folding and assembly of α-subunits (Matias et al. 2022; reference [5]).

  • Perhaps few phrases should be added into the introduction indicating what are proteasomes and their importance for the cells.

We thought that this may end up being a bit re­dundant in a special issue as tribute to Dr. Goldberg. However, the reviewer is right in that every paper has to stand alone. We thank the reviewer for pointing this out and have started the revised introduction with a section on the functions and importance of proteasomes.

Minor points:

  • Lane 204 contain extra text;

Has been removed. Thanks for spotting.

  • Lane 228 I could not find Figure S3C and D in Supplementary material;

Indeed, these panels had been taken out before submission. Thanks for noticing.

  • Western blots in general, lack positions of molecular weight markers;

Size markers have been added to the revised figures.

  • What is a CE Figure1,2,3, please indicate;

CE stands for crude extracts. This is now indicated in the figure legends.

  • Please indicate what is HMW and LMW in Figure legends;

Is now indicated in the revised figure legends.

  • The text has some typos.

We have tried to correct all typos

Again, thanks for spotting all these mistakes!

Reviewer 3 Report

Comments and Suggestions for Authors

This is a technically sound paper demonstrating the existence of alpha5-alpha6-alpha7-Pba3-Pba4 assembly intermediate in yeast. I do not follow this field in details and I had extremely hard time figuring out the novelty of the results. The problem stared at the Introduction where the authors did not explain what question they were asking and did not distinguish what was known and what was discovered in this paper. This left me with impression that this intermediate was already known in mammalian cells. Yeast is great model organism to dissect eukaryotic cell biology when work cannot be done in mammalian cells. But what is the point of simply repeating in yeast what is already known in mammalian cells? Mot of  discussion retells the results and does not integrate findings with what is known about assembly of mammalian proteasomes. In summary, this manuscript needs an extensive revision.

Minor point – there is a formatting problem in lane 204. Looks like “click to tap text here” should be removed.

Author Response

This is a technically sound paper demonstrating the existence of alpha5-alpha6-alpha7-Pba3-Pba4 assembly intermediate in yeast. I do not follow this field in details and I had extremely hard time figuring out the novelty of the results. The problem stared at the Introduction where the authors did not explain what question they were asking and did not distinguish what was known and what was discovered in this paper. This left me with impression that this intermediate was already known in mammalian cells. Yeast is great model organism to dissect eukaryotic cell biology when work cannot be done in mammalian cells. But what is the point of simply repeating in yeast what is already known in mammalian cells? Mot of  discussion retells the results and does not integrate findings with what is known about assembly of mammalian proteasomes. In summary, this manuscript needs an extensive revision.

We thank the reviewer for characterizing our paper as technically sound in demonstrating the existing of the a5-a6-a7-Pba3-Pba4 assembly intermediate in the yeast model system, and for pointing out that we have point out more clearly what our research question was and what are our novel findings as compared to what is already known from the mammalian cells. We have therefore modified the introduction to emphasize that the early steps in the assembly are largely in the dark. Experiments with knockdown of the Ump1 chaperone or beta-subunits in mammalian cells led to the identification of rings of α-subunits, which were associated with Pac1-Pac2 and Pac3. However, no such intermediates have been clearly demonstrated to be normal intermediates in wild-type cells. In yeast cells, no α-rings have been detected thus far, further raising doubts of these structures to by intermediates in an otherwise rather conserved assembly pathway. Earlier assembly intermediates with parts of α-rings such as α5-α6-α7 together with PAC3-PAC4 have indeed neither been characterized in yeast nor in human cells. Therefore, the α5-α6-α7-Pba3-Pba4 complex characterized in our work is indeed a novel assembly intermediate for any eukaryotic cell. In the introduction of the revised manuscript, we have clearly stated that our research question was to learn what the function of the Pba3-Pba4 is in early CP assembly and weather it directly cooperates with Pba1-Pba2 in forming assembly intermediates. The result of our work is that, at least in yeast cells, Pba3-Pba4 promotes formation of a starting unit with α-subunits 5, 6 and 7, and that the Pba1-Pba2 chaperone apparently does not bind to this starting unit, at least in the absence of additional factors. This finding was surprising because the binding pockets for the C-terminal HbYX motifs of Pba1 and Pba2 were expected to be already available between α5 and α6, or α6 and α7, respectively (figure S3). In conclusion, we believe that our results are by no means repetitions of experiments in yeast that were already done in mammals. We have revised our discussion in order to make it clearer what had been already known and what is novel.

Minor point – there is a formatting problem in lane 204. Looks like “click to tap text here” should be removed.

Has been removed. Thanks for spotting.

Round 2

Reviewer 2 Report

Comments and Suggestions for Authors

I would like to thank the authors for the carful addressing of my comments. I suppose that the manuscript could be published in the present form. At the same time, during the final preparation of the manuscript I would like to ask the authors to perform stylistic and spelling refinement of the text at Lanes 368, 410 and 412.

Author Response

I would like to thank the authors for the carful addressing of my comments. I suppose that the manuscript could be published in the present form. At the same time, during the final preparation of the manuscript I would like to ask the authors to perform stylistic and spelling refinement of the text at Lanes 368, 410 and 412.

We thank the reviewer for the positive feedback and for spotting additional mistakes in the indicated lanes, all of which have been corrected.

Reviewer 3 Report

Comments and Suggestions for Authors

I do not request any further changes.

Author Response

I do not request any further changes.

We thank the reviewer again for the time and efforts invested to help us improve our manuscript.